# Galectin-3 Is Associated with Cardiac Fibrosis and an Increased Risk of Sudden Death

**DOI:** 10.3390/cells12091218

**Published:** 2023-04-23

**Authors:** Mingma D. Sherpa, Swati D. Sonkawade, Vinesh Jonnala, Saraswati Pokharel, Mahyar Khazaeli, Yan Yatsynovich, Mohamad A. Kalot, Brian R. Weil, John M. Canty, Umesh C. Sharma

**Affiliations:** 1Division of Cardiology, Department of Medicine, Jacob’s School of Medicine and Biomedical Sciences, University at Buffalo, 875 Ellicott Street, Buffalo, NY 14203, USA; mdsherpa@buffalo.edu (M.D.S.);; 2Division of Thoracic Pathology and Oncology, Department of Pathology, Roswell Park Comprehensive Cancer Center, Buffalo, NY 14203, USA; 3Department of Pathology, Jacob’s School of Medicine and Biomedical Sciences, Buffalo, NY 14203, USA

**Keywords:** biomarkers, cardiac fibrosis, galectin-3, inflammation, mortality, risk stratification, sudden cardiac death

## Abstract

Background: Myocardial fibrosis is a common postmortem finding among individuals with Sudden Cardiac Death (SCD). Numerous in vivo and in vitro studies have shown that increased galectin-3 (gal3) expression into the myocardium is associated with higher incidence of fibrosis. Although elevated gal3 expression is linked with myocardial fibrosis, its role in predicting the risk of SCD is unknown. Methods: We reviewed the clinical datasets and post-mortem examination of 221 subjects who had died suddenly. We examined myocardial pathology including the extent of cardiac hypertrophy, fibrosis, and the degree of coronary atherosclerosis in these subjects. In a select group of SCD subjects, we studied myocardial gal3 and periostin expression using immunohistochemistry. To further examine if a higher level of circulating gal3 can be detected preceding sudden death, we measured serum gal3 in a porcine model of subtotal coronary artery ligation which shows an increased tendency to develop lethal cardiac arrhythmias, including ventricular tachycardia or fibrillation. Results: Of the total 1314 human subjects screened, 12.7% had SCD. Comparison of age-matched SCD with non-SCD subjects showed that SCD groups had excessive myocardial fibrosis involving both the left ventricular free wall and interventricular septum. In pigs with subtotal coronary artery ligation and SCD, we detected significantly elevated circulating gal3 levels approximately 10 days preceding the SCD event. Immunohistochemistry showed increased myocardial gal3 and periostin expression in pigs that died suddenly, compared to the controls. Conclusion: Our study shows that increased gal3 is associated with a higher risk of myocardial fibrosis and the risk of SCD. This supports the importance of larger translational studies to target gal3 to prevent cardiac fibrosis and attenuate the risk of SCD.

## 1. Introduction

Sudden cardiac death (SCD) from ventricular arrhythmia is a leading cause of mortality worldwide [1,2,3]. There are more than 356,000 out-of-hospital SCDs annually in the United States with nearly 90% being fatal [4]. Currently, there is a paucity of literature on the clinico-pathological determinants of SCD. Consequently, we lack definite SCD surveillance guidelines to monitor those at risk. Reduced left ventricular ejection fraction (LVEF) is currently the primary prognostic factor for predicting SCD, which may not be a sufficient approach since less than 20% of SCD is seen in those with severely reduced LV function (LVEF < 35%) [5]. Therefore, there is a strong need for developing more effective approaches to identifying risk-prone subjects. Prior studies demonstrated that the majority of SCD victims had abundant myocardial fibrosis in post-mortem analysis [6]. In fact, myocardial fibrosis is now considered as the most common finding associated with SCD in absence of other contributory causes [7,8]. We previously published the first data on the function of macrophage-expressed galectin-3 (gal3) in predicting advanced cardiac fibrosis [9].

Galectins are a family of beta-galactoside-binding proteins that share a consensus sequence in the carbohydrate recognition domain [10]. Among 17 galectin families that have been identified so far, gal3 is the most widely studied family member and can be found intracellular in the cytoplasm and nucleus, as well as extracellularly in various tissues [11,12,13]. Gal3 is secreted by macrophages, mast cells, other inflammatory cells, and stimulates cells to release various growth factors, as well as pro-inflammatory cytokines [14]. Gal3 also localizes into the cardiomyocytes immediately after the induction of myocardial ischemia [15]. We and several other groups previously documented a robust link between gal3 and cardiac fibrosis, which led to the approval for gal3 testing by enzyme-linked immunosorbent assays (ELISA) by the US Food and Drug Administration (FDA) [16,17,18].

In 2019, we reported data showing that circulating gal3 is associated with poor survival in resuscitated patients after out-of-hospital cardiac arrest. Gal3 enhanced the predictive utility when combined with other clinical covariates [19]. Studies have also examined numerous other factors including LV end-diastolic volume index, creatinine, arterial pH, serum lactate, length of resuscitation, type of rhythm at the time of arrest, and patient location at the time of events as the predictors of poor outcome [20,21,22]. However, there are limited data examining the predictive utility of both circulating and myocardial gal3 expression in myocardial fibrosis and SCD. In this study, we tested the hypothesis that increased circulating gal3 levels could precede sudden cardiac death. In subjects who died suddenly, we also examined the degree of cardiac fibrosis and myocardial gal3 expression, with a hypothesis that increased gal3 is associated with the risk of sudden death. Accurate risk stratification of SCD will facilitate earlier intervention and potentially improve post-cardiac arrest outcomes.

## 2. Methods

### 2.1. Human Post-Mortem Studies

#### 2.1.1. Study Design and Data Abstraction

The autopsy program at Kaleida Health, the local tertiary care hospital for the Buffalo Niagara Community, has an existing inventory of 1600 post-mortem specimens. We screened 1314 subjects that underwent comprehensive post-mortem analysis and identified 221 subjects that died suddenly. Of the 221 subjects, 167 deaths were reported as SCD. SCD was defined as non-traumatic, unexpected circulatory arrest occurring within 1 h of the onset of symptoms in an apparently asymptomatic subject. The controls included 54 subjects who died from causes other than SCD. We abstracted the clinical, laboratory, and imaging data from the hospital records of all subjects who met the inclusion criteria. The demographics included age, race, gender, body mass index (BMI), and family history of SCD. Clinical comorbidities included diabetes mellitus, hypertension, heart failure, coronary artery disease, obesity, and prior history of revascularization. Location of arrest (home/public/hospital), having an implantable cardioverter-defibrillator (ICD), length of cardiopulmonary resuscitation, and time to the return of spontaneous circulation (ROSC) were also included.

#### 2.1.2. Postmortem Procedures

Minimum standards required for assessment of sudden death in the general population were previously published [23]. All postmortem studies were structured examinations according to previously published guidelines [23]. After getting the relevant clinical information for the autopsy, an external examination of the body is done. Once a full autopsy is performed to exclude common and uncommon extra-cardiac causes of SD, the standard gross examination of the heart is conducted.

#### 2.1.3. Tissue Processing and Storage

A complete transverse (short-axis) transection of the heart at the papillary muscle insertion (mid-ventricular) level was made. Additionally, short-axis parallel slices of ventricles at 1 cm interval from the apex to the bases including subendocardial, mid-myocardial, and subepicardial myocardium were made. In cases of sudden death, a clinical autopsy was done to determine whether the death was due to cardiac disease or one of the many non-cardiac causes along with a clinical presentation on admission. Triphenyltetrazolium (TTC) staining is used in the autopsy room for preliminary diagnosis of acute myocardial infarction [24]. The heart is sliced transversally at a thickness of 1 mm and immersed in neutral TTC solution for 15 min to 20 min at 37 °C [25]. Once emptied of blood, total heart weight and wall thickness measurements were obtained, and various myocardial sections were stained with H & E [23]. Tissues were harvested, formalin-fixed, and paraffin-embedded. The histopathology slides are retained for at least 10 years from the time of tissue retrieval, following the guidelines set by the College of American Pathologists (CAP) [26]. Representative sections of the right ventricle, left ventricle, and septum were submitted for histological evaluation.

#### 2.1.4. Myocardial Tissue Morphometry and Immunohistochemistry

The extent of myocardial fibrosis was visualized by Masson’s trichrome staining (Thermo Scientific, Masson trichrome kit #22110648). Whole section images were obtained from Leica Aperio VERSA whole slide imaging System at 63× magnifications (Multispectral Imaging Suite, University at Buffalo). The total myocardial area and the area of positive staining for fibrosis were quantified using a color deconvolution algorithm that calculates the total stained area (percentage of weak, medium, and strong positive and negative staining) corresponding to the red and blue colors of the trichrome stain. The fibrosis percentage is assigned a numerical output using a validated algorithm reported previously [27,28]. The system was carefully programmed to evaluate fibrosis, highlighted in bright blue with Mallory trichrome staining in contrast to deep red-stained myocardium [28]. Visual and morphometric histological analyses at basal, midventricular, and apical levels were performed. Commonly, a score of 0 is given when fibrosis is absent. For any presence of fibrosis, score + is equal to ≤30% of the examined myocardium, score ++ when present in >30% to <60% of the examined myocardium, and score +++ when present in ≥60% of the examined myocardium [28]. In our study, the presence of any fibrosis was considered significant. The tissue was incubated with Anti-Galectin-3 (Abcam, Cambridge, MA; ab76245) and Anti-Periostin (Abcam, Cambridge, MA; ab215199) at a working dilution of 1 in 100, followed by Goat Anti-Rabbit IgG H&L (Alexa Fluor^®^ 594) secondary antibody at 1 in 250 dilutions. Nuclear DNA was labeled in blue with DAPI (Invitrogen, P36931). Images were taken with an epifluorescence microscope (Leica DMi8 inverted) at 100× magnifications.

### 2.2. Porcine Model

The animal care and experimental protocols followed US National Institutes of Health guidelines and were approved by the Institutional Animal Care and Use Committees of the University at Buffalo at New York.

#### 2.2.1. Experimental Porcine Model of SCD

The initial surgical preparation has been previously described [24,25] but modified in that proximal stenosis was placed on two of the three coronary arteries [23]. Briefly, juvenile pigs were premedicated with a Telazol (tiletamine 50 mg/mL and zolazepam 50 mg/mL)/ketamine (100 mg/mL) mixture (0.037 mL/kg im) and given prophylactic antibiotics (cephalothin 50 mg/kg iv and gentamicin 5 mg/kg im) [23]. A thoracotomy was performed under isoflurane anesthesia (1–3%). The proximal LAD and left circumflex coronary (LC) arteries were dissected free and instrumented with 1.5-mm (ID) Delra constrictors [25]. The right coronary artery (RCA) that supplies the posterior descending artery was not manipulated [23]. Pain was controlled with an analgesic (butorphanol 0.025 mg/kg im) and an intercostal nerve block (2% Marcaine) [23].

#### 2.2.2. Use of Porcine Model for Gradual Coronary Occlusion

The pig heart is most closely analogous to the human heart in terms of size, anatomy, and physiology [26,27]. The myocardium of a pig’s heart does not possess extensive collateral circulation similar to the human heart [28]. Because of this, the pig heart does not tolerate acute coronary occlusion well, but it can tolerate a gradual coronary occlusion. If a coronary artery is slowly occluded, it can be used as a model for chronic myocardial ischemia, chronic myocardial infarction, and heart failure [29]. Left ventricular systolic function was quantified using M mode fractional shortening (<25% was considered impaired) and ejection fraction was measured using Simpson’s biplane method, as described previously [30]. The porcine model developed a mildly reduced ejection fraction and had SCD within 1–4 months after coronary instrumentation. This clinical translation was important since the majority of SCA presents with severe, yet asymptomatic coronary artery disease (CAD) with minimal or no reduction of cardiac function.

#### 2.2.3. Myocardial Periostin Staining

Flash frozen 10μ thick pig heart LAD sections were fixed with 4% paraformaldehyde, followed by 0.1% Triton X-100 permeabilization and 1% BSA/10% normal goat serum tissue blocking. The tissue was incubated with Anti-Periostin (Abcam, Cambridge, MA; ab215199) at a working dilution of 1 in 100, followed by Goat Anti-Rabbit IgG H&L (Alexa Fluor^®^ 594) secondary antibody at 1 in 250 dilutions. Nuclear DNA was labelled in blue with DAPI (Invitrogen, P36931). Images were taken with an epifluorescence microscope (Leica DMi8 inverted) at 100× magnifications. Periostin is often expressed by cardiac myofibroblast and was used as a marker of a dynamic scar.

#### 2.2.4. Myocardial Collagen Content

The extent of total myocardial fibrosis was visualized by trichrome staining (Thermo Scientific™ Richard-Allan Scientific™ Masson Trichrome Kit #22110648). Whole section images were obtained from Leica Aperio VERSA whole slide imaging System at 63× magnifications (Multispectral Imaging Suite, University at Buffalo). The total myocardial area and the area of positive staining for fibrosis were quantified using color deconvolution algorithms. Briefly, whole section images were opened in Aperio ImageScope analysis software [v12.4.0.5043]. From the toolbar, the rectangular box was drawn around the tissue to define the area of interest, and a negative pen tool was used to remove peripheral fibrosis and unwanted tissue section. The pre-existing positive pixel count v9 algorithm was used for a color deconvolution to separate stains. Other parameters were set with manual scoring on a number of high-power fields and intensity thresholds. The algorithm calculates the total stained area (percentage of weak, medium, and strong positive and negative staining) corresponding to the red and blue colors of the trichrome stain. The fibrosis percentage was assigned a numerical output using a validated algorithm reported previously [31].

#### 2.2.5. Circulating and Myocardial Galectin-3 Measurements

For circulating gal3 measurement, we used a pre-coated gal3 (LSBio, Seattle, WA kit #LSF32161) specific 96 well strip microplates ELISA kit. In our quantitative enzymes-linked assays, Gal3 ELISA was performed in triplicate well using pig serum samples collected on baseline and post-instrumentation. This colorimetric quantitative sandwich ELISA measures the optical density, which was normalized with pre-determined standard gal3 concentration according to the manufacturer’s recommendation [32]. The porcine myocardial tissue was incubated with Anti-Galectin-3 (Abcam, Cambridge, MA; ab76245) at a working dilution of 1 in 100, followed by Goat Anti-Rabbit IgG H&L (Alexa Fluor^®^ 594) secondary antibody at 1 in 250 dilutions. Nuclear DNA was labeled in blue with DAPI (Invitrogen, P36931). Images were taken with an epifluorescence microscope (Leica DMi8 inverted) at 100× magnifications.

#### 2.2.6. Statistical Analyses

We analyzed the data using standard descriptive statistics along with mean and standard deviation for continuous variables and frequencies for categorical variables. We tested the association between the categorical variables and hard outcomes using Chi-square or Fisher exact test as appropriate. Data are expressed as mean ± standard deviation and percentage (%) for continuous and categorical variables, respectively. The odds ratio (OR) and 95% confidence interval (CI) were calculated for each independent variable with a significant *p*-value. Differences were considered to be significant when *p* < 0.05. Statistical data analysis was performed using SPSS version 23.0 (IBM Corp, Armonk, NY, USA, 2015).

## 3. Results

### 3.1. Clinical Characteristics

The mean age of the 221 subjects at the time of death was 62.4 ± 15.6 years. There were 59.7% whites, 33.9% blacks, 3.6% Hispanics, and 1.4% other races. Of 139 men, 81% of them had cardiac causes of death, whereas 19% had non-cardiac causes of death. The socio-demographic variables are shown in Table 1. In 82 females, 67% had cardiac causes of death, whereas 33% had non-cardiac causes of death. The mean body weight in our subjects was 90.6 ± 29.6 kg. The mean heart weight for an average 87-kg man ranges from 277 to 481 g [33]. In our study, subjects with sudden cardiac death had the mean heart weight of 549.2 ± 150.3 g while those with death from non-cardiac causes had a heart weight of 493.2 ± 103.8 g (*p* = 0.013). The mean LV thickness of the men who died from cardiac causes was 1.8 ± 0.4 cm, as compared with 1.6 ± 0.4 cm for those who died from non-cardiac causes (*p* = 0.005). Those who died from cardiac causes had higher heart weight, lower EF, increased LV thickness, and higher BMI as shown in Table 2. The place of arrest (hospital vs. home vs. public) did not show any significant differences. We have summarized the demographic, clinical characteristics, and circumstances surrounding SCD in Table 1.

Categorical variables in Table 3 show that left ventricular fibrosis (OR: 3.02, 95% CI: 1.62–5.61, *p* < 0.005), septal fibrosis (OR: 8.35, 95% CI: 2.71–26.37, *p* < 0.005), coronary atherosclerosis (OR: 4.05, 95% CI: 1.87–8.82, *p* < 0.005), acute myocardial infarction (OR: 3.27, 95% CI: 1.71—6.24, *p* < 0.005), and chest pain (OR: 3.98, 95% CI: 1.57—9.74, *p* < 0.005) were top five important factors in subjects who died from SCD. Patients with septal fibrosis (OR: 8.35, 95% CI: 2.71–26.37, *p* < 0.005) were eight-fold more likely to die suddenly from cardiac than non-cardiac causes. A total of 95% of subjects with septal fibrosis died from SCD in comparison to 84% of subjects with left ventricular fibrosis. The data for spatial fibrosis in relation to the region of the heart were derived from the autopsy report. The presence of coronary atherosclerosis was seen in 80% of SCD. By multivariate analysis, diabetes (OR: 2.59, 95% CI: 1.257–5.200, *p* = 0.028) and hypertension (OR: 2.69, 95% CI: 1.394–5.095, *p* = 0.013) were both independently associated with SCD.

### 3.2. Increased Myocardial Fibrosis and gal3 Expression in SCD Subjects

The results of Masson’s Trichrome-stained tissue sections (Figure 1A–C) demonstrated total percentage area of fibrosis was significantly higher in the sudden cardiac death subjects than in the non-cardiac sudden deaths (% Fibrosis: Non-SCD, 13.3% ± 3.4%; SCD, 19.3% ± 3.9%; *p* = 0.0383, *n* = 4–8). Within the same study group, we noted increased gal3 expression [gal3 (CTF): Non-SCD, 12,260 ± 769.2; SCD, 22,806.6 ± 2448.2; *p* = 0.0247, *n* = 4–6] (Figure 2B,D) and periostin expression [periostin (CTF): Non-SCD, 7706.2 ± 479.9; SCD, 9769.6 ± 431.9; *p* = 0.0345, *n* = 4–6] in myocardial autopsy specimen obtained from SCD subjects compared to controls (Non-SCD) group.

### 3.3. Increased Myocardial Fibrosis in a Porcine Model with SCD

These porcine models developed mildly reduced ejection fraction and had SCD within 1–3 months after coronary instrumentation with sub-total occlusion of LAD and left circumflex coronary vessels. To study myocardial fibrosis, we performed a quantitative evaluation of cardiac collagen content in Masson’s trichrome-stained tissue sections. Total percentage area of fibrosis was significantly higher in the porcine models of coronary ligations that died suddenly (% Fibrosis: Non-SCD, 12.01% ± 1.8%; SCD, 25.0% ± 4.09%; *p* = 0.0361, *n* = 5–6). The sudden death group also showed increased expression of periostin in the myocardium of the animals that died suddenly compared to controls [Periostin (CTF): Non-SCD, 8736 ± 629.5; SCD, 11,453 ± 888.2; *p* = 0.0463, *n* = 5–6] (Figure 3B,D).

### 3.4. Mild Reduction in Cardiac Function in a Porcine Model with SCD

The clinical data were obtained during follow-up at 3 and 4 months post coronary instrumentation to create a gradual coronary occlusion model similar to humans. The results in porcine models showed that LVEF and LVFS showed mean LVEF: 57.5 ± 3.5% and mean LVFS: 30.2 ± 3.5% at 3 months post-surgery vs. LVEF: 57.4 ± 9.0% and mean LVFS: 31.5 ± 4.7% at 4 months post-surgery. The results in controls (without any coronary instrumentation) showed a mean LVEF: 65.4 ± 4.3%. The results in controls (without any coronary instrumentation) showed a mean LVEF: 65.4 ± 4.3%. (Appendix A includes two avi files showing the mild reduction in EF based on the induced chronic ischemia created post-instrumentation.)

### 3.5. Increased Circulating and Myocardial Galectin-3 in Porcine Model with SCD

We found that higher gal3 levels were detected approximately 10 days prior to subjects with VT/VF cardiac arrest compared to 30–40 days before sudden cardiac death [Galectin-3 (pg/mL): survivors, 257.5 ± 18.83; SCD, 458.1 ± 22.8; *p* < 0.0001, *n* = 5–9] as seen in Figure 4A,B. Figure 4C,D are the representative immunofluorescence images of galectin-3-stained pig left ventricular sections with non-SCD and SCD. The bar diagram (Figure 4E) shows a significantly increased level of galectin-3 in these animals (*p* < 0.0197).

## 4. Discussion

Our study reports several novel observations. Firstly, we found that circulating gal3 plays a critical role in risk stratifying SCD by showing increased circulating gal3 levels 10 days prior to SCD as compared to 30 days. Similarly, immunohistochemical analysis showed increased periostin and trichrome (markers of fibrosis) in those with SCD. Unlike other clinical biomarkers, gal3 has pro-fibrotic activity and can lead to loss of cardiac function [9]. We have previously shown that in patients with acute myocardial infarction, circulating gal3 overexpression is associated with major adverse cardiovascular outcomes [32]. We characterized the role of gal3, both in vitro and in vivo, in our prior studies [9,31,32]. It is now important to recognize that sustained increased levels of circulating gal3 can be an aberrant mechanism during the transition from SCA to SCD. Continued surveillance of elevated gal3 levels can be a useful predictor in identifying the high-risk population for SCD. Likewise, increased serum and myocardial gal3 protein levels were noted in a translational preclinical model of coronary microembolism-induced acute myocardial injury [32]. In addition to the prognostic role of gal3 demonstrated in patients with ACS and coronary microembolism-induced acute MI model [32], our study adds to the field by demonstrating a new association between circulating gal3 levels and SCD. We have also shown that myocardial gal3 expression closely follows the expression of other molecules representing myocardial fibrosis, such as periostin.

Secondly, increased left ventricular and septal fibrosis is associated with increased mortality with SCD. There has been increasing evidence supporting the link between myocardial fibrosis and ventricular arrhythmia [22,34]. Both human subjects and the porcine model of SCD showed significant increase in percent fibrosis (Figure 1 and Figure 3). Cardiac fibroblasts have abundant intracytoplasmic and perinuclear gal3 receptors. Intrapericardial infusion of gal3 leads to LV dysfunction and excess collagen I deposition [9]. Nevertheless, we found RV fibrosis played a noncritical role in the pathogenesis of SCD. RV-free wall fibrosis may worsen paradoxical sepal motion, causing RV-LV dyssynchrony, and quite often may lead to supraventricular tachyarrhythmia, but malignant ventricular arrhythmias are rare [26]. Most prior studies done on myocardial interstitium and fibrosis have focused on LV. Therefore, the relevance of RV fibrosis to an arrhythmogenic substrate is limited and needs to be further investigated. In our study, 50% of SCD subjects had recorded arrhythmias (ventricular tachycardia, ventricular fibrillation, asystole, pulseless electrical activity) prior to arrest. Increased interstitial deposits of collagen filaments act as insulating barriers, supporting conduction slowing and conduction block [27]. Variations in cardiac conduction and conduction block, resulting in re-entrant wave front of excitation, are the classically recognized arrhythmic consequences of increased cardiac fibrosis [27].

Along with left ventricular fibrosis and septal fibrosis, other variables such as the presence of coronary atherosclerosis, acute myocardial infarction, higher BMI, increased heart weight, and LV thickness put subjects at higher risk for SCD. Gal3 was associated with increased left ventricular mass [29]. Other groups have reported marked up-regulation of gal3 in a pressure-overloaded heart, confirming hypertrophic cardiac remodeling [9,28]. Despite advances in the treatment and prevention of cardiovascular disease (CVD) and reduction in total cardiovascular mortality, the incidence of SCD remains high. Emerging evidence on gal3 may be able to help stratify patients who are at increased risk of SCD. Medical optimization of other comorbidities such as hypertension and diabetes along with regular follow-up will help lower the risk for SCD.

## 5. Study Limitations

This study has only a modest sample size as described above. Most of the patient population was 59.7% Caucasian, despite attempts to include a diverse patient population from multiple centers. Only 37% of subjects with SCD met the criteria. The staining was only performed on limited samples depending on the time of autopsy and the quality of the tissue specimen. The highest quality staining was obtained in these human subjects and the porcine model. Furthermore, one might point to the other variables that can affect survival in out-of-hospital cardiac arrest, including the medication provided during resuscitation and initiation of hypothermia protocol. Nevertheless, even after the adjustment for the majority of biomarkers and clinical variables, an elevated circulating gal3 level remains as a significant predictor of poor cardiovascular outcome in these subjects with SCD. The predictive role of circulating gal3 may be further enhanced when combined with other clinical variables, which may help in further risk stratification and guide management in these patients. Despite important advances over the last 10 years, ancillary examinations, such as toxicology, microbiology, and sampling for genetics are, all too often, not adequately performed to extensively study other factors involved in SCD [30].

## 6. Conclusions and Future Implication

Our results emphasize the importance of larger translational studies to assess the effects of galectin-3 for cardiac remodeling, fibrosis, and SCD. Additionally, similar to gal3, galectin-1 (gal1) and galectin-9 (gal9) are widely studied members of the galectin family. Prior studies also have shown the involvement of gal1 and gal9 in sudden death and heart failure. Studies have shown that gal1 promotes the development of atherosclerosis [35] and gal9 participates in the regulation of atherosclerosis [36]. In our future studies, we aim to continue our research with other galectin family members such as gal1 and gal9. Furthermore, the heterogeneity of SCD pathophysiology, where acquired and genetic aspects coexist in the development of fatal ventricular arrhythmias, can affect the strategies of risk prevention [37]. Current guideline recommendations do not emphasize the importance of advanced imaging techniques, including cardiac MRI, in the decision-making process of ICD implantation [38,39,40]. This aspect represents the greatest potential field for future research and clinical trials because of the high accuracy of MRI to identify endomyocardial fibrosis, which acts as a substrate for tachyarrhythmias [37]. Emerging biomarkers such as gal3 and advanced imaging-guided risk assessment may potentially change the landscape and decision-making in the primary prevention of SCD.

## Figures and Tables

**Figure 1 cells-12-01218-f001:**
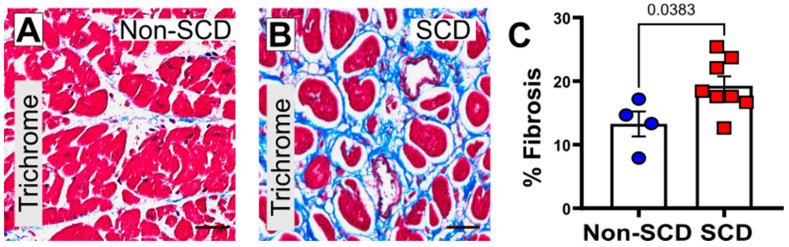
Comparison of myocardial fibrosis in control (non-SCD) vs. sudden cardiac death (SCD). Trichrome staining highlights collagen content (blue staining) in the LV myocardium in SCD and control subjects. (**A**,**B**) Representative images of the left ventricular sections from non-SCD and SCD subjects, respectively. (**C**) Quantitative analysis of myocardial collagen volume fraction. N = 4 (non-cardiac deaths) and 8 (sudden cardiac death). Scale bar: 50 µm, magnification ×400.

**Figure 2 cells-12-01218-f002:**
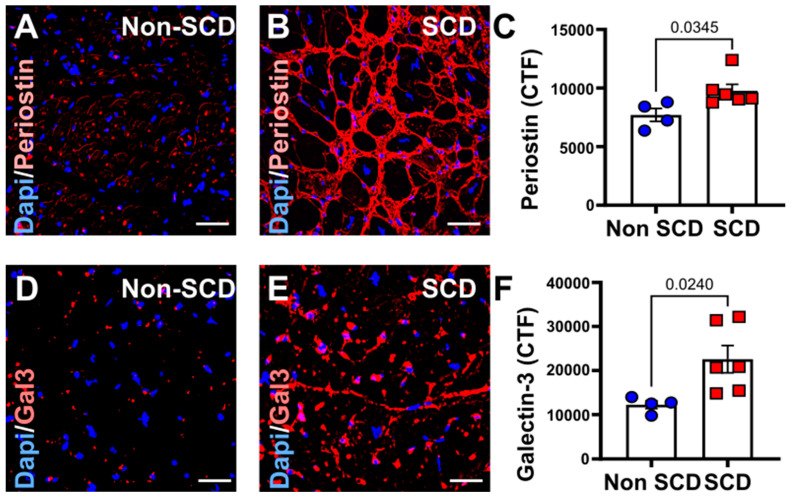
Comparison of myocardial periostin and galectin-3 in control (non-SCD) vs. sudden cardiac death (SCD) groups. Immunohistochemical analysis of formaldehyde-fixed, paraffin-embedded post-mortem human heart sections with periostin (**A**–**C**) and galectin-3 (**D**–**F**) in non-SCD vs. SCD. Increased periostin expression (*p* < 0.05) in myocardial autopsy specimen obtained from control (non-SCD) groups vs. SCD subjects shown in the bar diagram in the right upper panel. Similarly, increased galectin-3 expression (*p* < 0.05) in myocardial autopsy specimen obtained from control (non-SCD) groups and SCD subjects is shown in the bar diagram in the right lower panel. CTF, corrected total fluorescent intensity. Scale bar: 50 µm, magnification ×400.

**Figure 3 cells-12-01218-f003:**
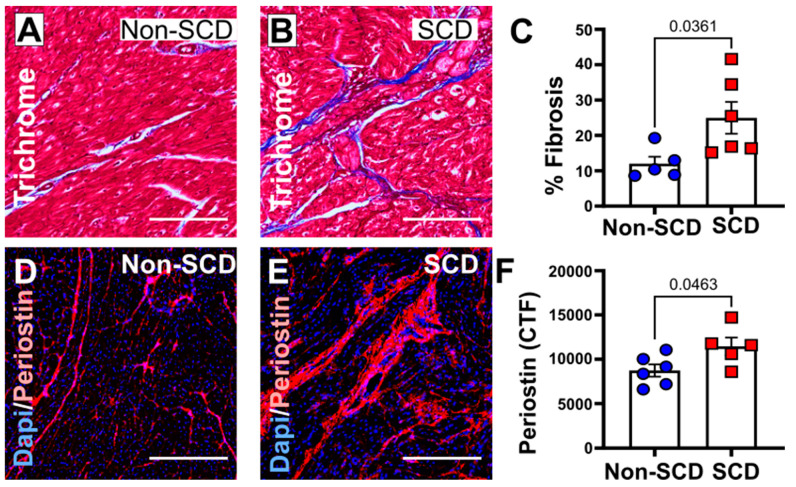
Comparison of myocardial fibrosis burden in porcine model of control (non-SCD) vs. sudden cardiac death (SCD). (**A**–**C**) Representative images of trichrome-stained left ventricular sections in pigs with non-SCD and SCD, respectively. Bar diagram in the right upper panel shows significantly increased collagen volume fraction in these animals. (**D**–**F**) Representative images of periostin immunohistochemical-stained left ventricular sections in pigs with non-SCD and SCD. Bar diagram in the right lower panel shows significantly increased periostin levels in these animals. N = 5–6 per group. CTF, corrected total fluorescent intensity. Scale bar: 200 µm, magnification ×100.

**Figure 4 cells-12-01218-f004:**
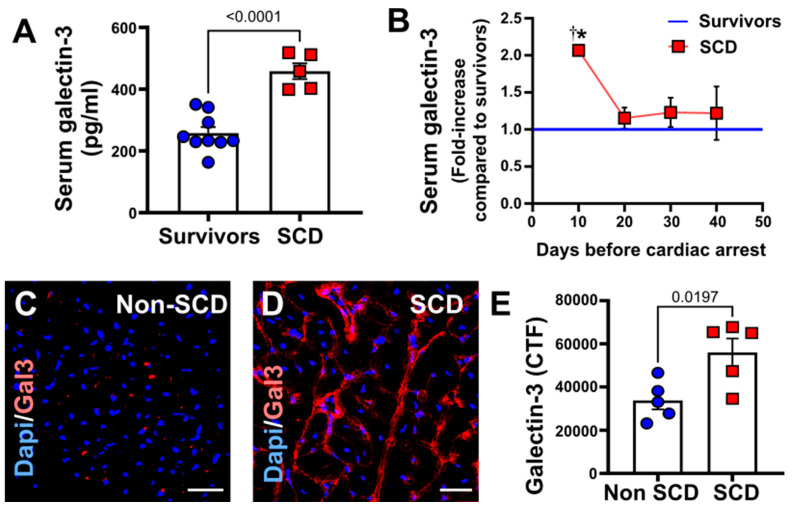
Comparison of circulating gal3 levels in porcine model (SCD vs. survivors). Panel (**A**) shows comparison of circulating galectin-3 levels in survivors (N = 9) vs. SCD group (N = 5) approximately 3 months after the coronary intervention. Panel (**B**) shows the time-course of circulating galectin-3 increase with higher levels detected ~10 days before ventricular tachycardia/ventricular fibrillation (VT/VF) cardiac arrest, compared to the mean galectin-3 levels measured in the pigs that had no cardiac arrest. N = 3–11, *, *p* = 0.0036 10 days before cardiac arrest vs. 20 days before cardiac arrest in SCD pigs, †, *p* = 0.0002 10 days before cardiac arrest in SCD vs. survivors pigs. (**C**,**D**) shows representative immunofluorescence images of galectin-3-stained pig left ventricular sections with non-SCD and SCD. The bar diagram (**E**) in the lower panel shows a significantly increased level of galectin-3 in these animals. N = 5 per group. CTF, corrected total fluorescent intensity. Scale bar: 50 µm, magnification ×400.

**Table 1 cells-12-01218-t001:** Socio-demographic and baseline characteristics of the study subjects who died suddenly from cardiac or non-cardiac causes along with concomitant clinical variables and cardiac tissue morphometry.

Categorical Variables		Non-Cardiac Cause of Death (*n* = 54)	Cardiac Cause of Death (*n* = 167)	Total	*p*-Value
				(*n* = 221)	
Sex	Female	27 (33%)	55 (67%)	82	0.024 *
	Male	27 (19%)	112 (81%)	139	
Family History of SCA		1 (5%)	21 (95%)	22	0.068
Chest pain before Arrest		6 (12%)	44 (88%)	50	0.001 *
NYHA Class	0	45 (28%)	113 (72%)	158	0.169
	1	2 (14%)	12 (86%)	14	
	2	1 (11%)	8 (89%)	9	
	3	1 (10%)	9 (90%)	10	
	4	0 (0%)	11 (100%)	11	
Revascularization		5 (10%)	43 (90%)	48	0.025 *
LBBB		1 (20%)	4 (80%)	5	0.480
RBBB		7 (37%)	12 (63%)	19	0.268
LV Fibrosis		20 (16%)	108 (84%)	128	0.001 *
RV Fibrosis		3 (19%)	13 (81%)	16	0.583
Septal Fibrosis		3 (5%)	55 (95%)	58	0.000 *
Coronary Atherosclerosis		37 (20%)	150 (80%)	187	0.000 *
Minimal/Mild CAD		15 (45%)	18 (55%)	33	0.000 *
Acute MI		15 (14%)	93 (86%)	108	0.000 *

Results in Table 1 are expressed as absolute numbers (percentage). *, *p* < 0.05. CAD, coronary atherosclerotic disease; CHF, congestive heart failure; LBBB, left bundle branch block; LV, left ventricle; MI, myocardial infarction; NYHA, New York Heart Association; RBBB, right bundle branch block; RV, right ventricle; SCA, sudden cardiac arrest.

**Table 2 cells-12-01218-t002:** Risk factors, clinical data, and significant laboratory values of the study subjects who died suddenly from cardiac or non-cardiac causes.

Continuous Variables	Non-Cardiac Cause of Death	Cardiac Cause of Death (*n* = 167)	Total	*p*-Value
	(*n* = 54)		(*n* = 221)	
Age	62.4 ± 17.8	62.3 ± 14.9	62.4 ± 15.6	0.972
Weight	100.2 ± 41.5	87.5 ± 24	90.6 ± 29.6	0.006 *
Height	1.7 ± 0.1	1.7 ± 0.1	1.7 ± 0.1	0.386
BMI	36.2 ± 15	31 ± 8	32.3 ± 10.4	0.001 *
BSA	2.1 ± 0.4	2 ± 0.3	2 ± 0.4	0.034
Ejection Fraction	57.3 ± 9.8	40.5 ± 18	42.9 ± 18.1	0.004 *
Systolic Blood Pressure	103.5 ± 40.3	95.5 ± 44.1	97.4 ± 43.2	0.392
Diastolic Blood Pressure	62.2 ± 33.6	55.3 ± 28.7	56.9 ± 29.9	0.292
Heart Rate	101.2 ± 25.5	84.6 ± 31.2	88.4 ± 30.7	0.013 *
Creatinine	2.4 ± 2.1	2.4 ± 2.2	2.4 ± 2.2	0.982
Troponin	0.9 ± 2	34.6 ± 153.1	26.3 ± 133.6	0.224
Myoglobin	1242.4 ± 2527	1342.3 ± 4151.2	1327.8 ± 3942.7	0.942
CK-MB	283 ± 753.2	100.5 ± 188.3	135.7 ± 367.5	0.140
LV Thickness	1.6 ± 0.4	1.8 ± 0.4	1.7 ± 0.4	0.005 *
RV Thickness	0.6 ± 0.7	0.6 ± 0.3	0.6 ± 0.4	0.668
Septal Thickness	1.5 ± 0.4	1.7 ± 0.5	1.7 ± 0.4	0.291
Mitral Valve	10.2 ± 1.2	10.3 ± 1.4	10.2 ± 1.4	0.681
Aortic Valve	7.1 ± 1.2	7.5 ± 1	7.4 ± 1	0.060
Tricuspid Valve	12.1 ± 1.5	11.9 ± 2.2	12 ± 2.1	0.642
Pulmonic Valve	8.2 ± 1.1	8.1 ± 1.1	8.1 ± 1.1	0.869
Heart Weight	493.2 ± 103.8	549.2 ± 150.3	535.9 ± 142.4	0.013

Results in Table 2 are expressed as mean ± SD, with comparisons made by student T-tests or Chi-squared tests, as appropriate. *, *p* < 0.05. BMI, body mass index; BNP, brain natriuretic peptide; BSA, body surface area; CK-MB, creatinine kinase-myoglobin binding; HDL, high-density lipoproteins; HTN, hypertension; ICD, implantable cardioverter defibrillator; LDL, low-density lipoprotein; LV, left ventricle, LVEF, left ventricular ejection fraction; MI, myocardial infarction; NYHA, New York Heart Association; PEA, pulseless electrical activity; RV, right ventricle.

**Table 3 cells-12-01218-t003:** Clinical variables and cardiac tissue morphometry data of the study subjects who died suddenly from cardiac or non-cardiac causes.

Categorical Variables	Non-Cardiac Cause of Death (*n* = 54)	Cardiac Cause of Death (*n* = 167)	Total (*n* = 221)	*p*-Value	Odds Ratio (Via Baptista-–Pike)	95% CI
Septal Fibrosis	3 (5%)	55 (95%)	58	0.000 *	8.35	2.706, 26.370
Coronary Atherosclerosis	37 (20%)	150 (80%)	187	0.000 *	4.05	1.869, 8.815
Minimal/Mild CAD	15 (45%)	18 (55%)	33	0.000 *	0.21	0.096, 0.470
Acute MI	15 (14%)	93 (86%)	108	0.000 *	3.27	1.707, 6.242
LV Fibrosis	20 (16%)	108 (84%)	128	0.001 *	3.02	1.626, 5.606
Aspirin	9 (11%)	75 (89%)	84	0.000 *	4.56	2.114, 10.200
Chest pain before Arrest	6 (12%)	44 (88%)	50	0.001 *	3.98	1.566, 9.743
Statin	11 (13%)	74 (87%)	85	0.002 *	3.50	1.632, 7.551
HTN	29 (19%)	124 (81%)	153	0.013 *	2.69	1.394, 5.095
Revascularization	5 (10%)	43 (90%)	48	0.025 *	3.45	1.326, 8.439
DM	12 (14%)	71 (86%)	83	0.028 *	2.59	1.257, 5.200

Results in Table 3 are expressed as absolute numbers (percentage), with comparisons made by student T-tests or Chi-squared tests, as appropriate. *, *p* < 0.05. CAD, coronary atherosclerotic disease; DM, diabetes mellitus; HDL, high-density lipoproteins; HTN, hypertension; LV, left ventricle; MI, myocardial infarction.

## Data Availability

The data presented in this study are available on request from the corresponding author. The data are not publicly available due to the sensitivity of patient information.

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
