# Peer review of "Galectin-3 Is Associated with Cardiac Fibrosis and an Increased Risk of Sudden Death"

_cells, 2023, doi:10.3390/cells12091218_

Round 1

Reviewer 1 Report

Authors investigated the role of gal3 in risk of myocardial fibrosis and onset of SCD. This is a highly relevant research area which conducted in human biopsies.

Major revision

·        Authors conducted preclinical evaluation in pig model to recaptitulate the SCD in human. However, conducted tissue morphology and morphometry studies relevant with galectin expression only. For explaining the myocardial function in this model, authors must include echo cardiography data/images to further prove the hypothesis.

·        Tissue storage and processing of human biopsies should be included

Minor revision

·        Minor typo errors found

·        Recent and relevant literature can appropriately cited

Author Response

Hi,

Thank you, reviewer 1, for your thoughtful reviews. Please see the below for a response. I have also highlighted ( with grey color in the response file) the changed/ new/ revised section in the manuscript itself (with yellow color). We have separated the Tables into parameters that are more pertinent to SCD so that it is clearer to the reader and created the supplemental table for other remaining data. Please let me know if there are any questions.  

Sincerely,

Responses to the Reviewers’ Comments

We are thankful for the opportunity to present a revised version of our manuscript titled "Galectin-3 is Associated with Cardiac Fibrosis and an Increased Risk of Sudden Death.” We would also like to sincerely thank all the reviewers for your thorough and in-depth evaluation of the manuscript and intriguing and insightful comments.

Reviewer #1

Comment 1: Authors conducted a preclinical evaluation in a pig model to recapitulate SCD in humans. However, conducted tissue morphology and morphometry studies are relevant to galectin expression only. To explain the myocardial function in this model, authors must include echocardiography data/images to further prove the hypothesis.

Response 1: Thank you for your insightful suggestions. The amended version now includes cine (movie) images showing the cardiac wall motion defects in pigs that underwent coronary intervention and were at risk of sudden death. The quantitative data outline left ventricular volume and function  have also been provided (method section 2.2.2 and 3.4).

Method: 2.2.2. Porcine Model for gradual coronary occlusion: Left ventricular systolic function was quantified using M mode fractional shortening (< 25% was considered impaired) and ejection fraction was measured using Simpson's biplane method, as described previously1.

Result: 3.4. Mild reduction in cardiac function in a porcine model with SCD: The clinical data were obtained during follow-up at 3 and 4 months post coronary instrumentation to create a gradual coronary occlusion model similar to humans. The results in porcine models showed that left ventricle ejection fraction (LVEF) and left ventricle fractional shortening (LVFS) showed mean LVEF:  57.5± 3.5% and mean LVFS: 30.2 ± 3.5 % at 3 months post-surgery versus  LVEF: 57.4± 9.0% and mean LVFS: 31.5 ± 4.7 % at 4 months post-surgery. The results in controls (without any coronary instrumentation) showed a mean LVEF: of 65.4± 4.3%.

Echocardiography: Video Echocardiography data of Normal Porcine versus Chronic Ischemic Porcine Model (attached as supplemental data).

Comment 2. Tissue storage and processing of human biopsies should be included.

Response 2: Thank you for this important comment. Based on your comment, we have added the following section 2.1.3. subtitled Tissue Processing and Storage.

Method: 2.1.3. Tissue Processing and Storage

A complete transverse (short-axis) transection of the heart at the papillary muscle insertion (mid-ventricular) level was made. Additionally, short-axis parallel slices of ventricles at 1.0 cm intervals from the apex to the bases including subendocardial, mid-myocardial, and subepicardial myocardium was made. In cases of sudden death, a clinical autopsy is done to determine whether the death was due to cardiac disease or one of the many non-cardiac causes along with a clinical presentation on admission. Triphenyltetrazolium (TTC) staining is used in the autopsy room for preliminary diagnosis of acute myocardial infarction2. The heart is sliced transversally at a thickness of 1 mm and immersed in a neutral TTC solution for 15 min to 20 min at 37°C3. Once emptied of blood, total heart weight and wall thickness measurements were obtained, and various myocardial sections were stained with H & E4.  Tissues were harvested, formalin-fixed, and paraffin-embedded. The histopathology slides are retained for at least 10 years from the time of tissue retrieval, following the guidelines set by the College of American Pathologists (CAP)5. Representative sections of the right ventricle left ventricle, and septum were submitted for histological evaluation.

Comment 3. Recent and relevant literature can be appropriately cited.

Response 3: This suggestion is highly appreciated. We have cited more recent and relevant literature to further strengthen our manuscript.

Reviewer 2 Report

In this study, Sherpa et.al investigated how is gal-3 associated with cardiac fibrosis and sudden cardiac death.Overall, this study has an interesting topic, good rationale and clear hypothesis, detailed method section and appropriate references. However, it can be further improved by addressing the comments below:

Major comments:

1.     Table 1-3 contains a lot of different parameters, it is hard to extract information. Maybe consider keeping the important and relevant parameters in the main table and move the rest to supplemental tables. Also, significant p- values can be highlighted and label with different numbers of *, and state this in figure legend. Also make sure to write full names for all of the abbreviations, for instance, LBBB etc. 

2.     Scale bars are missing in all tissue images.

3.     For IHC images, please color code the different staining, for instance, blue for DAPI and red for your protein staining.

4.     In table 2, there are LV, RV and Septal fibrosis, how did you define fibrosis, how many percent fibrosis or range do you consider these patients have fibrosis. Please clearly state this in method section. 

5.     Regarding LV fibrosis, in table 2, you have SCD 108 patients and 20 non-SCD patients, it is not clear why and how you picked 4 non-SCD and 8 SCD patients in figure 1.

6.     Are you using the same sample for figure 1 and figure 2? Why is there only 6 SCD samples in figure 2?

7.     Remake figure 4B, it is not clear what is the serum level of gal-3 in survivors at earlier time point.

8.     In the porcine models, does figure 3 and figure 4 using the same sample? If you are not using the same sample, it is not rigorous to link fibrosis with serum gal-3 levels. 

9.     Figure 4, only showed the serum level of gal-3, is there increased gal-3 in the myocardium? 

Minor comments:

1.     Typo, line 144, 10μ 

2.     How is the porcine model generated should also be stated briefly in text section 3.3 and why you want to use this model. 

Author Response

Hi,

Thank you, reviewer 2, for your thoughtful reviews. Please see the attached file for a response. I have also highlighted ( with grey color in the response file) the changed/ new/ revised section in the manuscript itself (with yellow color). We have separated the Tables into parameters that are more pertinent to SCD so that it is clearer to the reader and created the supplemental table for other remaining data. Please let me know if there are any questions.  

Sincerely,

Reviewer 3 Report

This manuscript authored by Sherpa, et al. indicates that galectin-3 (gal3) is associated with cardiac fibrosis risk of SCD. It may have translational implications for clinical practice. 

Major concerns

1. Tables 1, 2, and 3. "Results are expressed as mean +/- SD, absolute numbers (percentage), with comparisons were made by student T-tests or Chi-squared test, as appropriate." Authors did not present mean and SD in Tab. 1 and Tab. 3. Authors should clarify the data expression on individual table. 

2. ACEi was not indicated in the tables. Ace = ACEi in tables? 

3. Serum levels of gal3 were detected and expressed in porcine model of LAD. Expression of gal3 in heart tissue was not shown in porcine model. In comparison to porcine study, expression of gal3 in human heart tissue was presented in Fig. 2. but not serum levels of gal3 in humans. Can authors explain the study design and data presentation. 

4. Gal1 is also associated with heart inflammation and hypertrophy. It is recommended to measure other types of galectins also, at least gal1 and gal9. 

5. The statistical analysis between survivors and SCD in Fig. 5 must be provided.

Minor concerns.

None.

Author Response

Hi,

Thank you, reviewer 3, for your thoughtful reviews. Please see the attached file for a response. I have also highlighted ( with grey color in the response file) the changed/ new/ revised section in the manuscript itself (with yellow color). We have separated the Tables into parameters that are more pertinent to SCD so that it is clearer to the reader and created the supplemental table for other remaining data. Please let me know if there are any questions.  

Sincerely,

Round 2

Reviewer 1 Report

The authors incorporated substantial changes as suggested by the reviewers.

Reviewer 2 Report

The authors have addressed all of my comments.